# Persistence of *Metarhizium brunneum* (Ascomycota: Hypocreales) in the Soil Is Affected by Formulation Type as Shown by Strain-Specific DNA Markers

**DOI:** 10.3390/jof9020229

**Published:** 2023-02-09

**Authors:** Iker Hernández, Clara Sant, Raquel Martínez, Marta Almazán, Marta Caminal, Víctor Quero, Mohammed El-Adak, Albert Casanova, Inmaculada Garrido-Jurado, Meelad Yousef-Yousef, Enrique Quesada-Moraga, José Manuel Lara, Carolina Fernández

**Affiliations:** 1Futureco Bioscience, Avda. Del Cadí 19-23, 08799 Olèrdola, Spain; 2Department of Agronomy, ETSIAM, University of Córdoba, 14071 Córdoba, Spain

**Keywords:** *Bactrocera oleae*, entomopathogenic fungi, persistence, soil, molecular detection

## Abstract

The genus *Metarhizium* has an increasingly important role in the development of Integrated Pest Control against Tephritid fruit flies in aerial sprays targeting adults and soil treatments targeting preimaginals. Indeed, the soil is considered the main habitat and reservoir of *Metarhizium* spp., which may be a plant-beneficial microorganism due to its lifestyle as an endophyte and/or rhizosphere-competent fungus. This key role of *Metarhizium* spp. for eco-sustainable agriculture highlights the priority of developing proper monitoring tools not only to follow the presence of the fungus in the soil and to correlate it with its performance against Tephritid preimaginals but also for risk assessment studies for patenting and registering biocontrol strains. The present study aimed at understanding the population dynamics of *M. brunneum* strain EAMb 09/01-Su, which is a candidate strain for olive fruit fly *Bactrocera oleae* (Rossi, 1790) preimaginal control in the soil, when applied to the soil at the field using different formulations and propagules. For this, strain-specific DNA markers were developed and used to track the levels of EAMb 09/01-Su in the soil of 4 field trials. The fungus persists over 250 days in the soil, and the levels of the fungus remained higher when applied as an oil-dispersion formulation than when applied as a wettable powder or encapsulated microsclerotia. Peak concentrations of EAMb 09/01-Su depend on the exogenous input and weakly on environmental conditions. These results will help us to optimize the application patterns and perform accurate risk assessments during further development of this and other entomopathogenic fungus-based bioinsecticides.

## 1. Introduction

The genus *Metarhizium* is a soil fungus with worldwide distribution including more than 50 described species that are insect and mite entomopathogens playing a significant role in the control of many agricultural pests [1]. Of particular interest are the different approaches based upon the use of *Metarhizium* spp. to control Tephritid fruit flies [2,3,4,5,6,7]. Among them, it arises due to its efficacy and environmental safety the strategy based on soil treatments with *Metarhizium brunneum* targeting tephritid preimaginals in the soil, which has been shown as a key pillar of the control of the olive fruit fly *Bactrocera oleae* (Rossi, 1790), the primary pest of the olive crop [4,6,8].

The application of *Metarhizium brunneum* (Petch, 1935) (Hypocreales: Clavipitaceae) targeting Tephritidae preimaginals has been demonstrated to be an effective and environmentally friendly control method [3,4,6,7,9,10,11]. Particularly, *Metarhizium brunneum* strain EAMb 09/01-Su (herein, EAMb 09/01-Su) is an EPF with proven bioinsecticide activity against the olive fruit fly [5,7,12]. Physical contact between EPF and the target insect is required for the former to exert its bioinsecticide activity, so the life cycle stages of the olive fruit fly with restricted and accessible localization are the most suitable for treatments with EPF. For instance, 3rd instar larvae that fall to the ground to pupate, and adults emerging from pupae are suitable targets for EPF treatment, while flying adults or eggs, larvae, and pupae within the olives are not [4,11,13]. Thus, for the use of EPF-based insecticides against olive fruit flies it is of capital importance to understand the behavior of the EPF in the environment and how it fits in the pest and crop life cycles to achieve high and reproducible efficacies and accurate risk assessments.

Aside from the inherent infectivity of EPF, the formulation of mycoinsecticides is key to the success of these treatments (reviewed by Burges [14]). Co-formulants such as sunscreens, humectants, emulsifiers, stickers, etc. can promote physical contact between EPF and the target insect, enhance infectivity, and/or protect the EPF from harsh environmental conditions (see, for instance [15,16,17,18]). Furthermore, co-formulants can also confer properties such as extended shelf life or compatibility with conventional application machinery, which improve the marketability of EPF-based commercial products.

In this context, monitoring the fate, behavior, and population dynamics of particular *M. brunneum* strains released as biocontrol agents in the soil is a key challenge not only to follow the presence of the fungus in the soil and to correlate it with its performance against *B. oleae* preimaginals, but also for risk assessment studies for patenting and registering these biocontrol strains. Up to now, *M. brunneum* population studies in the soil have been based on the use of selective or semi-selective agar media [19,20] for dilution-plating soil samples according to the colony forming units (CFU) method. Therefore, it is not possible to distinguish between species and strains or to quantify total *Metarhizium* spp. propagules.

Since Bischoff et al. [21] established new species within the *Metarhizium* genus using the translation elongation factor 1-alpha (TEF 1-α) gene, efforts have been focused on the development of new molecular tools to distinguish these new species. A technique based on the nuclear ribosomal internal transcribed spacer (ITS) region, which is considered the standard barcoding marker for species-level identification in fungi [22], was developed for specific detection and quantification of *Metarhizium* clade 1 that contains *M. brunneum* [23,24]. A more precise species-specific multiplexed PCR assay has allowed us to identify and distinguish within the PARB clade species *M. pingshaense*, *M. anisopliae*, *M. robertsii*, and *M. brunneum* [25]. Up to now, the studies working on the presence, persistence, and diversity of *M. brunneum* have used all these previous tools, but these are unable to identify a specific *M. brunneum* strain or to distinguish *M. brunneum* from other species from the PARB clade.

In this scenario, the present study aimed at determining the persistence and population dynamics of *M. brunneum* strain EAMb 09/01-Su in the field when applied as a bio-insecticide. For this, different experimental formulations were applied in the field directly to the soil, where most of the olive fruit fly developmental stages are accessible and show restricted localization occurrence, and the levels of EAMb 09/01-Su were tracked using a strain-specific real-time qPCR method created after obtaining the whole genome shotgun (WGS) sequence of this strain.

## 2. Materials and Methods

### 2.1. Fungal Strain

The strain EAMb 09/01-Su belongs to the culture collection of the Agricultural Entomology Research Group AGR 163, from the Department of Agronomy (DAUCO) of the University of Cordoba. The strain is deposited in the Spanish Collection of Culture Types (CECT) with accession number CECT 20784.

### 2.2. Whole Genome Shotgun Sequencing and Assembly

EAMb 09/01-Su was grown in PDA plates at 26 °C for 2 days to obtain fungal biomass with no sporulation. Then, hyphal biomass was recovered from the plate and used for DNA extraction with the Quick DNA fungal/bacterial miniprep kit (Zymo Research, Irvine, CA, USA). The DNA was submitted to a service provider (IGATech) for sequencing. After initial quality control, the samples were used to prepare Truseq (PCR-free) sequencing libraries according to the manufacturer’s instructions (Illumina, Cambridge, UK), which were sequenced in an Illumna Novaseq6000 platform at 2 × 150 paired-end read mode.

The EAMb 09/01-Su genome was assembled using the RECONSTRUCTOR pipeline [26] by Sequentia Biotech. In short, raw reads were trimmed and mapped to the reference genome (*Metarhizium brunneum* ARSEF 3297; GenBank accession nr. GCF_000814965.1). After this first mapping, an iterative variant calling (IVC) was carried out: first, a variant calling is performed, next, the reference genome is modified according to the small variants and misassembles detected, and, finally, the raw reads are mapped again against the modified reference genome. This three-step procedure (mapping, variant calling, and reference genome modification) is iterated until the number of variants appearing stabilizes at low values.

The sequences that did not align with the (modified) ARSEF 3297 genome sequence were assembled de novo using SPAdes [27]. The resulting scaffolds were filtered out to keep only scaffolds > 2 kbp long with significant homology to Ascomycota sequences (GenBank) to remove erroneous assemblies and contaminants. The assembled scaffolds were included in the IVC genome assembly, and the raw reads were mapped again to test if they could provide missing links between scaffolds. None of the de novo assembled scaffolds could bridge the reference scaffolds, so the final EAMb 09/01-Su genome assembly consisted of the reference-assisted assembled scaffolds plus the de novo assembled scaffolds. This Whole Genome Shotgun project has been deposited at DDBJ/ENA/GenBank under the accession JALMGC000000000. The version described in this paper is version JALMGC010000000. The quality of the assembly was tested with BUSCO and QUAST [28,29].

### 2.3. Marker Design and Validation

From the de novo assembled scaffolds, which are specific to strain EAMb 09/01-Su as compared to *M. brunneum* strain ARSEF 3297, the workflow described by [30] for bacteria was applied to design and validate qPCR markers. The specificity of the primers and probes was verified against all fungal sequences deposited in the GenBank using Primer Blast [31]. Primer pairs and/or probes with possible off-targets according to PrimerBlast were discarded. Then EAMb 09/01-Su cultures were used as a template for real-time quantitative PCR (qPCR) using the primer pairs and labeled probes designed as described before. EAMb 09/01-Su was grown in PDA plates as described before and the biomass was recovered in sterile distilled water. An aliquot (100 µL) of the solution was heated for 5 min at 98 °C and kept on ice until used as qPCR template. Serial dilutions of the remaining volume were spread in PDA and grown as described before to determine the fungal concentration. qPCR reactions consisted of 1 µL each of three target mixes (a mixture of the two primers at 10 µM each and the corresponding dual-labelled probe at 4 µM), 10 µL PerfeCTa qPCR ToughMix (Quantabio, Beverly, MA, USA), and 7 µL template (boiled hyphae). The thermal cycling conditions were 95 °C 3 min, 45× (95 °C 15 s, 60 °C 1 min) and 10 °C forever. The detection, including the choice of probe labels, lamp settings, etc., was performed as recommended by the thermal cycler manufacturer (Light Cycler 480II, Roche Life Science, Basel, Switzerland). To validate the specificity of the markers (i.e., the combination of primer pair and probe), 26 *Metarhizium* spp. strains from the in-house Futureco Bioscience collection (Appendix A), along with the EAMb 09/01-Su strain, were grown in PDA, boiled and amplified as described before.

### 2.4. Preparation of Prototypes

Conidia were prepared by growing EAMb 09/01-Su in 2 stages. First, an EAMb 09/01-Su pre-inoculum at 5 × 10^6^ conidia·mL^−1^ was inoculated in 100 mL glucose-yeast extract liquid medium [32] and grown at 26 °C for 72 h with orbital shaking at 200 rpm. The resulting culture was inoculated to sterile vaporized rice spread over aluminum shelves at 0.25 g·cm^−2^ by adding 0.1 mL of culture and 0.9 mL saline solution (0.9% aqueous NaCl) per gram of rice. The mixture was kept for 10 days at 26 °C and ca. 100% relative humidity, and then transferred to a dry incubator (FD 53L Binder, Tuttlingen, Germany) for 24–72 h. After drying, the spores were separated from the rice grains using a fine mesh. To produce the oil dispersion (OD) prototype, adequate amounts of carrier, emulsifier, disperser, rheological agent, and conidia were added in a sterile glass beaker and homogenized with a disperser equipped with a Cowles-type propeller for 3 h at room temperature. To produce the wettable powder (WP) prototype, adequate amounts of carrier, UV filter, wetting agent, anticaking agent, rheological agent, dispersant, and conidia were added to a laboratory scale V-type mixer for 10 min. The concentration of conidia in the technical-grade active ingredient (i.e., the conidia) or the prototypes was determined by the plate-dilution method in PDA plates grown for 3 days at 26 °C.

To produce microsclerotia, 5 × 10^6^ conidia·g^−1^ were inoculated to 100 mL of the medium described previously by Mascarin, et al. [33] in 300 mL Erlenmeyer flasks and grown for 7 days at 28 °C with orbital shaking at 250 rpm. At the end of the culture period, microsclerotia were counted microscopically using a counting chamber. The viability of microsclerotia was assessed by spreading a known volume of microsclerotia culture in agar water plates (2% agar), growing the culture for 3 days at 26 °C, and counting the germinated microsclerotia. The fitness of the microsclerotia was assessed similarly but the agar plates were left to grow for 10 days and the spores were harvested in a nominal volume of 0.5% Tween80 and counted in a counting chamber [34]. The microsclerotia culture was mixed with clay and anti-caking in a ratio of 2% and 0.5% of the culture volume respectively. After 30 min of orbital shaking at 200 rpm, the mixture was vacuum filtered in a Buchner funnel using Whatman No 42 filter paper. The filter cake was air-dried for at least 48 h at room temperature. Then the cake was milled using a pestle and mortar and mixed with the carrier using a laboratory scale V-type mixer for 10 min.

### 2.5. Field Trial Design and Implementation

Four field trials were performed to track the persistence of EAMb 09/01-Su in commercial settings. In two of the field trials, 3 prototypes were studied, while in the other two trials, only the OD prototype was monitored. All trials consisted of 2 randomized blocks per treatment. Table 1 summarizes the main features of each trial.

The plots at Mont-Roig del Camp and Avinyó Nou were each divided into 8 sub-plots of ca. 1250/1000 m^2^ each (Mont-Roig del Camp and Avinyó Nou, respectively), with 30 trees in each sub-plot. Two sub-plots were assigned to each of the 4 treatments (untreated control, WP, OD, or MS). The applications of the prototypes were performed with a quad towing trailer carrying an auto-propelled 200-Ldeposit with integrated mixing and a hose connected to a gardening sprayer. The tank contained a solution of prototypes at 1% (*v*:*v* or *w*:*v*, depending on the prototype). The spraying was performed at 2 bars and it was aimed at the soil under the tree canopies (ca. 10 m^2^·tree^−1^) at a rate of 1.5 L·tree^−1^. Liquid samples of the sprayed liquid were collected directly from the nozzle when approximately half of each treatment had been completed to verify the EAMb 09/01-Su concentration. The applications were carried out twice: one at the end of Autumn, and another one at the end of Winter. Further details on the trial, including the doses and the application dates, are shown in Table 1. Aside from watering (the plots were equipped with drip irrigation system), the soil of the plots was not disturbed. The soils at these trials were both clay loams with pH 7.5 but differed in the total organic C content (1.05% in Mont-roig del Camp and 0.50% in Avinyó Nou).

In the trials at Caldes de Montbui, the experimental design was two blocks per treatment and 2 treatments (OD and untreated control). In these trials, the blocks were separated by 3 rows of untreated trees. The applications were carried out with a tractor equipped with an auto-propelled 600-L tank connected to an herbicide application bar with 2 110° nozzles 50 cm apart. The tank contained a solution of a prototype at 1% (*v*:*v*). The herbicide bar was placed at a height of 40–50 cm and the spraying was aimed at the soil under the tree canopies (ca. 1.5 m and 2 m to each side of the row in the super-intensive and traditional regime, respectively). Liquid samples of the sprayed liquid were collected directly from the nozzles when approximately half of each treatment had been completed to verify the EAMb 09/01-Su concentration. As in the trials at Mont-Roig del Camp and Avinyó Nou, the applications were carried out twice: one at the end of Autumn, and another one at the end of Winter. Again, further details on the trial, including the doses and the application dates, are shown in Table 1. The soil at Caldes de Montbui was the same for the two trials: a loam with pH 8.4 and 2.51% total organic C.

It is to be noted that since the treatments were applied to the soil under the tree canopies, depending on the plantation density, the use of border tree rows, the size of the trees, and canopy management, the percentage of the plot surface treated varied from ca. 25% in super-intensive plots (Caldes de Montbui in super-intensive regime) to ca. 50% in extensive plots.

### 2.6. Soil Sampling

For each experimental block, 3 soil samples were collected; each sample consisted of 4 pooled sub-samples, each drawn from a cardinal point 50 cm away from the trunk of a tree. Sub-samples consisted of 20–30 cm deep soil cores, extracted using an Auger probe of 2 cm inner diameter.

### 2.7. DNA Extraction from Soil Samples

Different commercial kits were tested to extract DNA from the different soils, and the best for each soil, based on the yield and the A_260_/A_280_ and A_260_/A_230_ ratios, was selected for sample processing. For the soils from Mont-Roig del Camp, the DNA was extracted with the Quick-DNA Fecal/Soil Microbe Miniprep™ (Zymo Research, Irvine, CA, USA) according to the manufacturer’s instructions using a Star-Beater ball mill (VWR, Radnor, PA, USA). The DNA from the samples from Avinyó Nou was extracted with the DNeasy Powersoil kit (Qiagen, Venlo, The Netherlands), and those from Caldes de Montbui (I and II), with the DNeasy Powersoil Pro kit (Qiagen), always according to manufacturer’s instructions using the Star-Beater ball mill too. All samples were dried overnight at 60 °C (FD 53L Binder) prior to DNA extraction to prevent differences due to the soil water content.

### 2.8. Strain Quantification from Field Samples

Calibration standards were prepared as follows. Soil samples from control plots before the beginning of the experiments were taken. The soil samples were pooled, dried at 60 °C for 24 h, and separated in 0.25-g aliquots. The aliquots were spiked with 20 µL of an EAMb 09/01-Su solution at increasing concentration, from 0 to 10^8^–10^9^ CFU·mL^−1^. The DNA was extracted from the mixtures and used for qPCR amplification as described before. A calibration standard was built from the threshold cycles (C_T_) and the spiked EAMb 09/01-Su concentration. From the calibration standards, the limit of detection (LOD; the minimal concentration at which a C_T_ is recorded, higher than that of the no-template control) and the limit of quantification (LOQ; the lowest concentration detected and included in the calibration curve) were obtained.

Once calibration standards were built, the DNA from the samples was qPCR-amplified as described before and the concentration of EAMb 09/01-Su was inferred from the calibration standard with the obtained C_T_’s. All calibrations and quantifications were performed with the LightCycler480 software (Roche, Basel, Switzerland).

Comparison of EAMb 09/01-Su concentrations during time and after the different treatments, was carried out applying ANOVA. Student’s *t*-tests were applied to compare pairs of selected data points.

### 2.9. Weather Data

Weather data were kindly provided by the Catalonian Meteorology Service (*Servei Meteorològic de Catalunya*) through the network of automated weather stations (*Xarxa d’Estacions Meteorològiques Automàtiques*) [35]. The EAMb 09/01-Su levels in each trial, after the treatment with each prototype, and determined with each marker, were filtered to discard all observations below the LOQ. After verifying that the data did not fit a normal distribution (Shapiro-Wilk’s test), the filtered data were analyzed by Spearman’s rank correlation method. The environmental parameters, together with the fungal concentration, were submitted to principal component analysis (PCA) using R Studio.

## 3. Results

### 3.1. Whole Genome Shotgun Sequencing

RECONSTRUCTOR took 6 iterations to assemble the EAMb 09/01-Su based on the reference genome. As shown in Table 2, the assembly of the EAMb 09/01-Su is slightly (3.3%) larger than the reference genome (Table 2).

This assembly is comprised of significantly more contigs due to the inclusion of 189 contigs assembled from unmapped reads, but the N50, the length of the longest scaffold, and the proportion of N’s are almost the same in both the reconstructed and the reference genome (Table 2). On the other hand, the completeness analysis shows that the reconstructed genome is slightly more complete than the reference genome since, from the 290 single-copy genes analyzed, the EAMb 09/01-Su assembly shows 286 genes completely assembled in a single copy, while the reference genome shows 285 (Table 2). The EAMb 09/01-Su whole genome shotgun sequence has been deposited in the DDBJ/ENA/GenBank under the accession JALMGC000000000.

### 3.2. Marker Design and Validation

From the contigs assembled using unmapped reads, 100 primer pairs with their corresponding internal hybridization oligos were designed in silico. Then, from these 100 primer/probe sets, 22 primer pairs were empirically tested for specificity towards EAMb 09/01-Su with intercalating dye detection chemistry against 26 *Metarhizium* spp. strains from the in-house Futureco Bioscience collection (Appendix A). Only 3 primer pairs showed complete specificity towards the intended target strain (Table 3 and Appendix A).

The complete molecular markers—primer pair plus the corresponding hydrolysis probe—were also specific to the intended target strain: qPCR reactions using hydrolysis probe detection chemistry showed no amplification with any of the off-targets tested (Appendix A).

### 3.3. Persistence of EAMb 09/01-Su in the Soil of Field Trials

#### 3.3.1. Mont-Roig del Camp

Soil samples from Mont-Roig del Camp were taken up to 244 days after the first application (DAA) and the parameters of the calibration curves are shown in Appendix A. Immediately after the first application, the levels of EAMb 09/01-Su raised from not detected to 6.3–7.7 × 10^3^ CFU·g^−1^ (depending on the marker selected) when applied as WP, and to 1.7–3.8 × 10^4^ when applied as OD (Figure 1).

These levels remained stable, with oscillations, until the second application (i.e., 92 DAA) (Figure 1). After the second application, the levels of the fungus grew progressively to reach their maximum 123 DAA, which was about 1.5 × 10^6^ and 5.0 × 10^5^ CFU·g^−1^ in the OD- and WP-treated soil samples, respectively (Figure 1). From 123 DAA until the end of the experiment the levels of EAMb 09/01-Su in soils treated with the WP prototype decreased about an order of magnitude, while in soils treated with the OD prototype EAMb 09/01-Su levels remained constant (Figure 1).

On the other hand, the EAMb 09/01-Su levels in the soil samples treated with MS showed more extreme values. Immediately after the first application of MS, EAMb 09/01-Su was only detected with the FAM- and R610-labeled markers (2.1 × 10^3^ and 1.9 × 10^3^ CFU·g^−1^, respectively), while it remained below the LOD (5.1 × 10^2^ CFU·g^−1^) in the CY5-labeled marker (Figure 1). These concentrations remained constant during the first 14 days and then dropped below the LOD in the FAM- and R610-labeled markers too. After the second application, the quantification of EAMb 09/01-Su showed inconsistent patterns in the three markers: the FAM-labeled marker showed an increment up to 1 × 10^8^ CFU·g^−1^ by day 97 DAA and then fell below the LOD, the CY5-labeled marker was only detected from 97 to 160 DAA (at about 10^8^ CFU·g^−1^), and the R610-labeled marker was only detected in a single sampling point (97 DAA, at 1.7 × 10^8^ CFU·g^−1^) (Figure 1). It is to be noted that the LOD increased significantly after the first 97 days of experiments, from 1.6 × 10^3^ to 8.5 × 10^7^ CFU·g^−1^ depending on the marker selected. At least one standard from the calibration curve must be included in each qPCR plate to normalize data across qPCR plates. Due to a large number of plates run, two calibration curves had to be prepared: one was used for samples until 97 DAA and the second for samples from 114 DAA.

Overall, when applied as OD and WP the strain EAMb 09/01-Su shows similar behavior, although with consistently higher levels in the OD- than in the WP-treated samples (*p* < 0.05, *t*-test), especially from the second application on (Figure 1). The population dynamic of the MS treatment is too inconsistent to define any trend.

The weather data showed weak, yet significant in some cases, correlation with the levels of EAMb 09/01-Su. In all cases, including those in which it was found to be statistically significant, the correlation coefficient was below |0.282| except for the MS measured with the CY5-labelled marker (Appendix A). In this case, EAMb 09/01-Su showed a clearly higher correlation with daily temperatures and relative humidity (ρ = |0.756|). This correlation, however, is to be considered with caution since the number of observations above the LOQ was very low (8) and was distributed on only two sampling dates.

#### 3.3.2. Avinyó Nou

Soil samples from the field trial at Avinyó Nou were harvested until 240 DAA (Figure 2) and the parameters of the calibration curves are shown in Appendix A.

According to the CY5- and R610-labeled markers, the dynamics of the EAMb 09/01-Su population when treated with the WP and OD prototypes was similar as at Mont-Roig del Camp: an initial increment, stabilization until the second application, another increment after the second application that led to highest EAMb 09/01-Su levels (about 1.1 × 10^6^ and 1.5 × 10^4^ CFU·g^−1^, in the OD- and WP-treated groves, respectively) by 107 DAA, and a progressive decline until the end of the experiment (Figure 2). The levels of EAMb 09/01-Su in groves treated with MS were below the LOD in all samples analyzed, except for one sampling point at 107 DAA where FAM-labeled marker attains 1.1 × 10^6^ CFU·g^−1^. Overall, as it occurred at Mont-Roig del Camp, the samples from Avinyó Nou treated with the OD prototype showed higher levels than those treated with the WP prototype (*p* < 0.05, *t*-test) of EAMb 09/01-Su throughout the experimental period (Figure 2).

The weather data showed weak, yet significant for some parameters, which correlate with the levels of EAMb 09/01-Su (in all cases ρ < |0.491; Appendix A). The relative humidity showed a mild but significant negative correlation with the EAMb 09/01-Su levels when applied to WP and OD formulations (ρ ≈ −0.3; Appendix A). On the other hand, the solar radiation showed a mild positive correlation (statistically significant, too) with the EAMb 09/01-Su levels when applied as WP and OD formulation (ρ ≈ −0.4; Appendix A). These correlations are not observed in samples from groves treated with MS (Appendix A).

#### 3.3.3. Caldes de Montbui I: Super-Intensive Regime

Soil samples from Caldes de Montbui under a super-intensive regime were taken up to 279 DAA (i.e., 179 DAB). The parameters defining the calibration curves are shown in Appendix A. After the first application of OD, the levels of EAMb 09/01-Su remained undetectable until the second application (100 DAA). Then, the levels of EAMb 09/01-Su raised, reaching a maximum of about 3.5 × 10^3^ CFU·g^−1^ (depending on the marker) by 115 DAA (Figure 3).

Then, the population of the fungus declined progressively, losing about an order of magnitude, until the end of the experiment (Figure 3; the R610-labeled marker lost traceability after day 157).

The weather data showed weak, yet significant in some cases, correlation with the levels of EAMb 09/01-Su (in all cases ρ < |0.418|; Appendix A). These correlations were negative with the temperatures, the rainfall, and the solar radiation but were positive with the relative humidity (Appendix A).

#### 3.3.4. Caldes de Montbui II: Traditional Regime

Soil samples from Caldes de Montbui under a traditional regime were taken up to 256 DAA (i.e., 156 DAB). The parameters defining the calibration curves are shown in Appendix A. EAMb 09/01-Su remained below the LOD at most sampling points. In the few samples in which EAMb 09/01-Su was detected—in all cases after the second application—it remained below the LOQ so no population dynamic trend could be identified (Figure 4).

## 4. Discussion

The EAMb 09/01-Su genome sequence assembly was of slightly better quality than the reference genome (*M. brunneum* ARSEF 3297) (Table 2). In addition, the EAMb 09/01-Su genome includes 189 contigs assembled de novo from reads that do not map the reference genome. Re-sequencing the EAMb 09/01-Su genome with a long-read platform may help us to improve the contiguity of the assembly, but the assembly obtained hereby allowed us to obtain strain-specific sequences to design qPCR assays for EAMb 09/01-Su detection.

Strain-specific markers for real-time PCR have proven to be a viable, affordable, and replicable method for DNA detection in different matrices [30,36,37,38,39,40]. In the present study, the markers designed to detect the presence of EAMb 09/01-Su DNA in soil samples from olive groves accomplished the quality and sensitivity criteria required for such a purpose. It is worth noting that 19 out of the 22 (86%) primer pairs tested amplified unintended targets even though they were predicted to be specific in silico. This implies that the sequence databases used for specificity testing in silico are hardly representative of the genome sequences present in the environment. Nevertheless, the absence of signal in non-inoculated soil samples confirmed that the qPCR assays described hereby were specific towards EAMb 09/01-Su, at least in the background of the local biodiversity. The calibration curve parameters and the LOD and LOQ values were variable depending on the soil source and prototype, but, except for MS, these values were in line with other previous studies using strain-specific qPCR (e.g., [30,39,40,41]).

Based on the EAMb 09/01-Su concentration in the prototype, the volume of broth applied, the surface treated, and the soil depth sampled, the concentrations are expected to be above the LODs in all trials. Still, there were clear differences in the detection of EAMb 09/01-Su that cannot be ascribed to the surface-based dose or the soil type. For instance, the fungus is not detected after the first application in any of the trials at Caldes de Montbui, and it is not detected after the second application in the soil at Caldes de Montbui under traditional management with the CY5- or R610-labeled markers. Thus, other factors than the dose or the soil type must have an impact on the EAMb 09/01-Su levels in the soil. The methodology for the detection of EAMb 09/01-Su applied as MS shows inconsistent results. On one hand, the three DNA markers were detected at different time points. On the other hand, the fungus was only detected in a few sampling points, and not necessarily immediately after the applications. It has been recently reported that microsclerotia from *M. brunneum* strains EAMa 01/58-Su in a clay loam kept at field capacity and 25 °C in the dark show peak production of infective propagules 50–70 days after inoculation [42]. The time points at which EAMb 09/01-Su, applied as MS, was detected do not match this timing after the first or the second application. The inconsistency of the EAMb 09/01-Su quantification, when applied as MS, may be due to an insufficient or inefficient sampling effort, together with an irregular distribution of the MS in the soil.

Overall, the exogenously applied EAMb 09/01-Su persisted during the ca. 250 days that lasted the trials at Mont-Roig del Camp and Avinyó Nou. In the trials at Caldes de Montbiu, in contrast, small differences in the LODs and LOQs lead to loss of EAMb 09/01-Su track at many time points and with the different markers since the population of the fungus is close to the LODs. Nevertheless, the levels of EAMb 09/01-Su several weeks after the last applications were about the same concentrations reported for *Metarhizium* spp. in other soils (e.g., [23,43,44]). Thus, the introduced EAMb 09/91-Su is able to establish itself in the soil, but it does not seem to behave as an invasive strain.

The peak of EAMb 09/01-Su concentration, and the range of variation in the population, were greater after the second application, which was carried out in late winter or early spring, when the weather conditions are more favorable for fungal development. This suggests that there might be an effect of the environment on the levels of the fungus. However, although EAMb 09/01-Su showed a moderate correlation with the solar radiation (ρ ≈ 0.4) and the relative humidity (ρ ≈ −0.3) in the trial at Avinyó Nou, these correlations were not consistent with other locations. A PCA analysis also reflects this inconsistency (Appendix A). On the other hand, the concentration of EAMb 09/01-Su in the fields treated with MS did not show consistent (i.e., backed up with the three markers) correlation with any of the environmental parameters tested.

The soil treated with the OD prototype showed consistently higher EAMb 09/01-Su concentrations than when treated with WP, and reached higher peak concentrations, too. Wettable powder and oil dispersion are formulation types suitable for active ingredients that are sensitive to water. Particularly, spore-forming microorganisms such as EAMb 09/01-Su may resume their metabolic activity and abandon their latent, resistant, state (the spores) upon increased water availability and/or temperature, thereby becoming more sensitive to subsequent harsh conditions. Among these two formulation types, oil dispersions have been reported to improve penetration and retention, particularly on hydrophobic surfaces [45,46,47,48,49,50,51,52]. These ends were not tested for the prototypes evaluated in the present study, but they may explain the better performance of the OD prototype as compared to the WP.

## 5. Conclusions

Overall, the present study shows that *M. brunneum* strain EAMb 09/01-Su can be tracked in soil samples by using strain-specific qPCR assays. These assays work as expected for OD and WP formulation prototypes, but further improvement is required to apply this method to MS formulations. Aside from the methodological contribution, it is hereby shown that EAMb 09/01-Su is able to establish in the soil when inoculated, at concentrations comparable to those shown in soils in which this species is native, and also to other related fungi.

## Figures and Tables

**Figure 1 jof-09-00229-f001:**
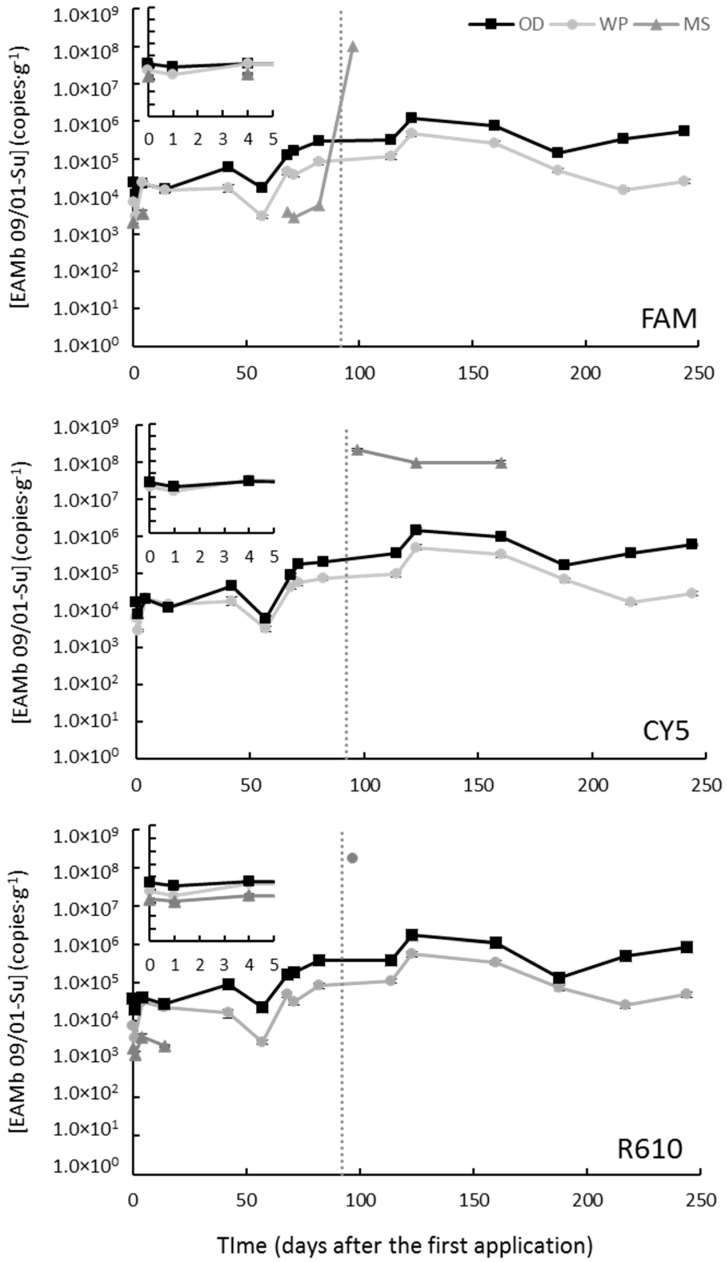
Concentrations of *Metarhizium brunneum* strain EAMb 09/01-Su in the substrate when applied as different formulations in a field trial at Mont-Roig del Camp. The different panes correspond to the three different markers, labeled as denoted within each pane. Missing data are measurements below the limit of detection. The vertical dotted line indicates the time of the second application, and the inserts show the first 5 days after the first application in detail. FAM stands for 6-carboxyfluorescein; CY5, for cyanine-5; and R610 for LightCycler^®^ Red 610.

**Figure 2 jof-09-00229-f002:**
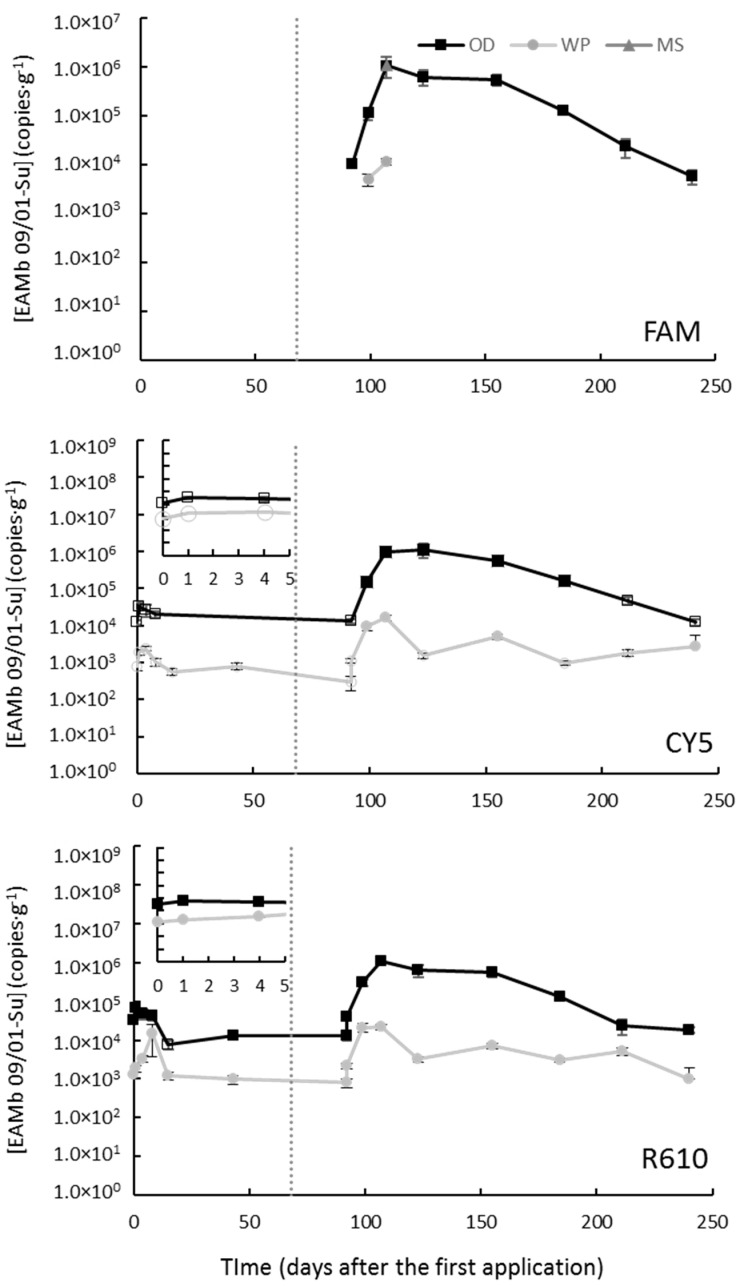
Concentrations of *Metarhizium brunneum* strain EAMb 09/01-Su in the substrate when applied as different formulations in a field trial at Avinyó Nou. The different panes correspond to the three different markers, labeled as denoted within each pane. Hollow symbols stand for values above the limit of detection but below the limit of quantification, and missing data are measurements below the limit of detection. The vertical dotted line indicates the time of the second application, and the inserts show the first 5 days after the first application in detail. FAM stands for 6-carboxyfluorescein; CY5, for cyanine-5; and R610 for LightCycler^®^ Red 610.

**Figure 3 jof-09-00229-f003:**
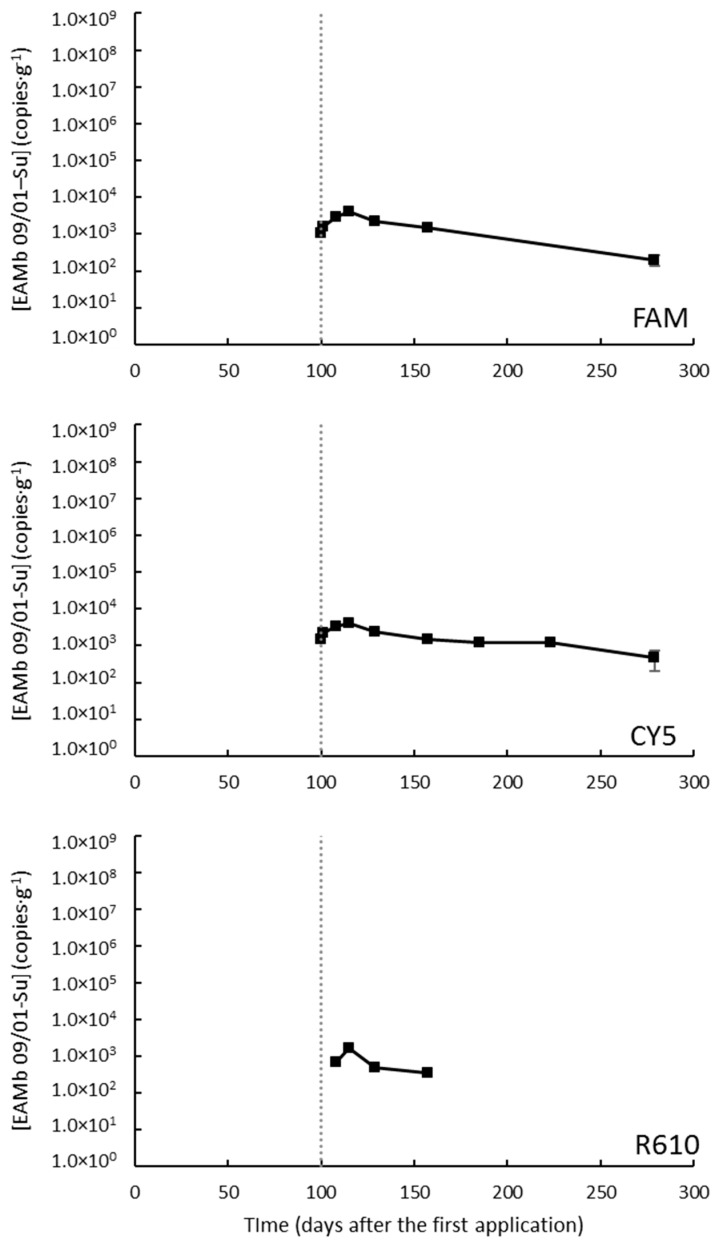
Concentrations of *Metarhizium brunneum* strain EAMb 09/01-Su in the substrate when applied as oil dispersion in a field trial at Caldes de Montbui under a super-intensive regime. The different panes correspond to the three different markers, labeled as denoted within each pane. Missing data are measurements below the limit of detection. The vertical dotted line indicates the time of the second application. FAM stands for 6-carboxyfluorescein; CY5, for cyanine-5; and R610 for LightCycler^®^ Red 610.

**Figure 4 jof-09-00229-f004:**
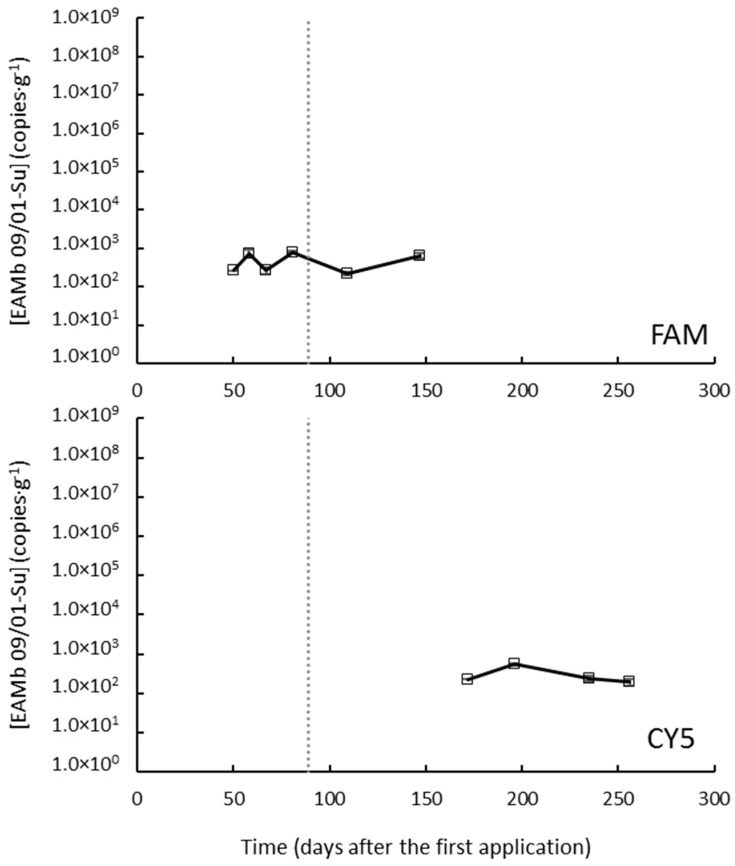
Concentrations of *Metarhizium brunneum* strain EAMb 09/01-Su in the substrate when applied as oil dispersion in a field trial at Caldes de Montbui under the traditional regime. The different panes correspond to different markers, labeled as denoted within each pane; EAMb 09/01-Su was not detected with the R610-labeled marker at any sampling point. Hollow symbols (in this case, all data points) stand for values above the limit of detection but below the limit of quantification, and missing data are measurements below the limit of detection. The vertical dotted line indicates the time of the second application. FAM stands for 6-carboxyfluorescein; CY5, for cyanine-5.

**Table 1 jof-09-00229-t001:** Main features of field trials.

	Mont-Roig del Camp	Avinyó Nou	Caldes de Montbui I	Caldes de Montbui II
Coordinates	41.0530, 0.95679	41.37614, 1.77810	41.63020, 2.15113	41.63066, 2.14766
Variety	Biancolilla	Picual	Arbequina	Vera del Vallès
Tree spacing (m × m)	6 × 7	6 × 6	1.5 × 4	5 × 7–7 × 7 ^1^
Prototypes tested	WP ^2^, OD ^3^ and MS ^4^	WP, OD and MS	OD	OD
Treated/untreated area (ha)	0.25 ^5^/0.25 ^5^	0.2 ^5^/0.2 ^5^	1.66 ^5^/1.09 ^5^	1.72 ^6^/1.72 ^6^
Date A ^7^ (mm/dd/yyyy)	25 November 2019	25 November 2019	17 November 2020	17 November 2020
Concentration A (%; *v*:*v*)	1	1	1	1
Volume A (L·ha^−1^)	1500 ^8^	1500 ^8^	500	500
Dose A (CFU·ha^−1^; ×10^10^)	11.9, 22.5, 187.5 ^9^	13.5, 31.5, 375 ^9^	4.0–4.6 ^1^	3.4
Date B ^10^ (mm/dd/yyyy)	25 February 2020	4 February 2020	25 February 2021	9 February 2021
Concentration B (%; *v*:*v*)	1	1	1	1
Volume B (L·ha^−1^)	1500 ^8^	1500 ^8^	500	500
Dose B (CFU·ha^−1;^ ×10^10^)	4.8, 4.35, 243^4^	2.94, 3.0, 195 ^4^	1075–3645	340–360
Machinery ^11^	Quad; 200-L auto-propelled deposit with integrated mixing; hose with a standard gardening nozzle		Tractor; 600-L auto-propelled deposit with integrated mixing; herbicide application bar with 2 nozzles (110°) 50 cm apart, 40–50 cm above the soil	

^1^ Depending on the plot. ^2^ WP stands for wettable powder, ^3^ OD, oil dispersion, and ^4^ MS, microsclerotia. ^5^ Divided into 2 plots of similar area. ^6^ Divided into 4 plots of similar area; ^7^ A denotes the first application. ^8^ 1.5 L·tree^−1^. ^9^ WP, OD, and MS, respectively. ^10^ B denotes the second application. ^11^ Images of the machinery used are shown in Appendix A.

**Table 2 jof-09-00229-t002:** Comparison of the assembly quality metrics for the reconstructed genome and the reference genome.

Statistic	EAMb 09/01-Su	ARSEF 3297
Total length (bp)	38,285,473	37,066,166
Contig number	278	92
Longest scaffold (bp)	7,146,780	7,151,295
N50	1,825,093	1,825,569
N’s per 100 kbp	223.06	264.87
Complete and single copy	286	285
Complete and duplicated	4	5
Fragmented	0	0
Missing	0	0

**Table 3 jof-09-00229-t003:** *Metarhizium brunneum* EAMb 09/01-Su strain-specific markers, including primers and probes.

Marker	Oligo	Sequence (5′→3′)	5′ Label	3′ Label
H977_1	FWD primer	CGTAGTAGTCGCGGGCTATC	N/A	N/A
	RCP primer	GCTCCAATGCCTCCGTAATA	N/A	N/A
	Probe	CCTGCCCAACCATCCATCCA	FAM	BHQ-1
H977_9	FWD primer	GGCATCAGAGCACATGAAGA	N/A	N/A
	RCP primer	GCCACGCCTCTAGAACAAAG	N/A	N/A
	Probe	TGTGGGTCACCGCGTCCAAA	CY5	BBQ
H977_12	FWD primer	AGTGGTGGATGGCAAAGTTC	N/A	N/A
	RCP primer	CAGCGCGTTATTTGTGCTTA	N/A	N/A
	Probe	CGGCACGGTCAACTGCTCCC	R610	BHQ-2

FAM, 6-carboxyfluorescein; CY5, cyanine-5; R610, LightCycler^®^ Red 610; BHQ, Black Hole Quencher^®^; BBQ, BlackBerry Quencher^®^-650.

## Data Availability

The Whole Genome Shotgun sequence of *M. brunneum* strain EAMb 09/01-Su has been deposited at DDBJ/ENA/GenBank under the accession JALMGC000000000. The version described in this paper is version JALMGC010000000.

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
