# Peer review of "Persistence of Metarhizium brunneum (Ascomycota: Hypocreales) in the Soil Is Affected by Formulation Type as Shown by Strain-Specific DNA Markers"

_jof, 2023, doi:10.3390/jof9020229_

Round 1

Reviewer 1 Report

This is an interesting manuscript about determining the persistence and population dynamics of Metarhizium brunneum strain EAMb 09/01-Su in the field when applied to the soil of 4 field trials using different formulations for the control of olive fruit fly, Bactrocera oleae, using strain-specific DNA markers after obtaining the whole genome shotgun (WGS) sequence of this strain.

The present work was organized logically, and the results obtained were reliable and persuasive. The results are well presented, their interpretation is relevant, and the methods are highly detailed. I would therefore recommend accepting this manuscript.

Author Response

No comments.

Reviewer 2 Report

In this manuscript the authors report studies on the longterm persistence in soil of a Metarhizium brunneum tested in the form of three formulations – a wettable, powder, an oil dispersion and a aqueous suspension microsclerotia.

I have no concerns about the molecular aspects of the study, namely the identification and preparation of strain specific DNA markers. Sufficient details are given not only for a reader to understand what was done, but also for someone else to replicate the methods. I also commend the use of commercial application methods to simulate a level of realism. And it is great the studies were carried out over two and not just one season and in 4 fields.

I do have a concern, a major concern, about the accuracy of DNA probes to detect ONLY live propagules in the soil samples. Conidia can be dead yet their DNA can be detected as if they were still alive and infectious. The DNA measurement methods are very sensitive and would include target DNA sequences as long as they were not denatured. There was no validation that the DNA of dead conidia was excluded from detection. The calibrations reported were only with fresh conidial suspensions, a calibration necessary in its own right. The ideal validation would be to determine DNA levels in comparison with colony forming unit determinations of “bad” conidial preparations (at least <50% viability, if not close to 0%) versus “good” preparations (viability >90%). Only by demonstrating that DNA of dead spores, hyphal fragments, microsclerotia are not detected, would the observations reported here be valid. That is only good science. There is a dilemma about creating dead conidia without denaturing the DNA so probes would not amplify it. High heat (50-60 C) for an hour or two would kill the spores but might denature the DNA. I do note that the soil samples were subjected to 60 C overnight… Formaldehyde and related would readily kill the spores but also might denature the DNA. My own method has been subjecting the dry spores to 45 C for several days to 2 weeks, by which time the conidia are largely dead. I do know that Metarhizium microsclerotia left at room temperature and allowed to equilibrate to typical ambient humidity (50-70%) will lose their viability within a month or so (a disadvantage of microsclerotia, requiring them to be extremely dry (<3% moisture and vacuum packed, for decent shelf life). I am also very concerned that a control treatment -- of conidia in simple aqueous suspension -- was not incorporated into the study. So the reader cannot be sure that it was the formulations that were responsible. (Microsclerotia are a separate issue. They are "apples and oranges" compared to the WP and OD, because microsclerotia are stabilized, melanized hyphae that must "germinate" and produce conidia for any efficacy.) 

Of secondary concern is the nature of the soil sampling. As I interpret the application descriptions these were both applications to the soil surface without direct incorporation. At 1500 L ha-1 that volume reduces to 0.015 ml mm^2 of soil surface, barely wetting the surface. There are a number of papers, going back to the 1970s, relating to poor soil percolation of conidial suspension. Only in very sandy soils do conidia in suspension percolate beyond a few cm and that is with considerable (many cm of overhead water application) watering in. Yet the soil cores were made to a depth of 20-30 cm, resulting in a considerable volume of soil with little probability of containing conidia, or hyphal fragments (assuming the conidia germinated and produced mycelium, which is really unlikely in nonsterile soils), thus even CFU determinations would be low-balled estimates. One has to sample the soil where the conidia are most likely to be present, regardless of formulation. With the application methods described, that sampling should have been done in the top 2-5 cm of the soil profile. In addition it has been my (sad) experience that soil applications result in a very heterogenous distribution of spores in the soil, so that a 2 cm probe is really a hit or miss affair, requiring at least 10-12 sub-samples per plot for any reasonable accuracy.

Of tertiary concern is that the formulation components were not described in more detail. Of course I realize these were probably EcoFutur’s proprietary formulation recipes, and thus proprietary. However, as it stands, the findings (ignoring my previous criticisms) are not readily applicable to parallel studies by others. I would also caution that my own experience with working with emulsifiable oil formulations in soil, is that the oil phase (in which the conidia are contained) can bind conidia to the soil and otherwise create very heterogenous distribution in soil applications.

Lastly, the miscrosclerotia were sprayed as suspensions? They are typically 50-200 microns in size and while I can see they could have been applied with garden nozzle without clogging, I am a bit amazed they were sprayed with regular agricultural hydraulic nozzles unless the nozzles were of very wide apperture, e.g. (American) Teejet 11005 or -6 or equivalent.

All that written I must sadly recommend rejection of the manuscript, at least until the verification studies I mentioned earlier have been completed.

Author Response

The authors are sorry to hear that the referee felt compelled to reject the manuscript. Anyway, his valuable comments have been oncorporated into the manuscriot, and responses to his/her comments are provided in the attached file. 

Reviewer 3 Report

The manuscript submitted by Hernández et al. reports a novel methodology to detect the persistence of a Metarhizium brunneum strain-specific in the soil of olive groves after different formulations of the strain were applied to four groves for control of fruit fly preimaginals. They assembled the genome of their candidate strain named EAMb 09/01-Su and found the strain-specific DNA markers to design paired primers for the detection of the candidate strain's DNA derived from the time-course soil samples post-application through real-time quantitative PCR analysis. Some of the designed primers were shown to effectively detect the strain-specific DNA in soil samples after the oil dispersion and wettable powder formulations of the candidate strain were applied. Their data reveal that the fungus persisted for up to 250 days in the soil. The whole study is an output from a large amount of field and laboratory work. The developed methodology helps to improve the application strategy of fungal formulations against the pest and is potential for use in risk assessment of applied fungal insecticides. The manuscript was written quite well but still needs a revision for improved clarity and conciseness.

Main suggestions:

1.      In Subsection 2.5, the field trials at four locations should be described in more details. Table 1 is a good summary but insufficient to understand the details. By the way, Table 1 needs a modification for clarity. Check up if the used tables follow up the journal's format.

2.     It is unclear why the same methodology failed to detect the strain-specific DNA in soil samples after the encapsulated microsclerotia formulation was applied. This needs an expanded discussion. Was this formulation unable to colonize the soil after application?

3.      The sentence "oil-dispersion formulation provides greater protection than wettable powder or encapsulated microsclerotia" is present in Abstract. It is not clear here whether 'greater protection' mean 'better control efficacy against the pest 'better cololnization or longer persistence in the soil'? Since the field trials were aimed at the control of fruit fly preimaginals, it would be good to see some data on the field control efficacies of different formulations or mention the mail result in this manuscript if the fungal efficacies against the pest will be presented in a separate paper.

Author Response

The authors thank the referee for his/her valuable comments and corrections. They manuscript has been modified accordingly. Please, find a point-by-point response in the attached file.

Reviewer 4 Report

Dear authors, the manuscript is well done and an interesting work.  You will find some minors suggestions and corrections in the attached file.

Regards,

Author Response

The authros thank the reviewer for his/her valuable comments. The manuscript has been corrected according to them. Please, find a poin-by-point response in the attached manuscript. Since the referee provided the comments in a PDF file,In this file, the authors refer to each comment in the attached document using the line counts of the original manuscript.
